# The Effect of Antibiotics on Bacteriome of *Sitophilus oryzae* and *Rhyzopertha dominica* as a Factor Determining the Success of Foraging: A Chance for Antibiotic Therapy in Grain Stores

Olga Kosewska, Sebastian Wojciech Przemieniecki *[ID] and Mariusz Nietupski [ID]

Department of Entomology, Phytopathology and Molecular Diagnostics, University of Warmia and Mazury in Olsztyn, Prawocheńskiego 17, 10-720 Olsztyn, Poland
*   Correspondence: sebastian.przemieniecki@uwm.edu.pl

**Featured Application:** **The sterilization of the internal microbiome of insects only causes the death of *S. oryzae*, and has no impact on *R. dominica*. The removal of the symbiotic microbiota causes a severe shift in the biochemical potential of the insect. Confirmation that certain insects enter into symbiotic interactions with the microbiota suggests the need for extensive testing of the pest's susceptibility to the loss of symbiotic microbiota, which could open a new path in warehouse pest control strategies.**

**Abstract:** Rice weevil (*Sitophilus oryzae*) and the lesser grain borer (*Rhyzopertha dominica*) are very important warehouse pests and, therefore, their control is crucial. At a key moment in the life of adult *Sitophilus* spp., the obligatory symbiotic nature of the *Sodalis pierantonius* bacterium opens up a new perspective for natural antibiotics and bactericides. In this study, we used nanopore sequencing for 16S rRNA barcoding to evaluate the internal bacteriome of *S. oryzae* and *R. dominica* and sterilized the insects' internal microbiome with gentamicin. The treatment of the interior of *S. oryzae* with gentamicin (30 mg·g$^{-1}$) hampered insect functioning (supposed lack of DOPA (4-dihydroxyphenylalanine) synthesis, stabilizing the exoskeleton by *Sodalis pierantonius* symbiont) and elicited a lethal effect in the first stages of this pest's adult life. In addition, we identified biochemical biomarkers (enzymatic activity and substrate utilization) active in living individuals, but inactive in dead individuals (e.g., C8 esterase/lipase and α-chymotrypsin).

**Keywords:** symbiotic microbiome; storage pests; nanopore sequencing; biochemical activities

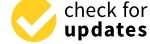



## 1. Introduction

The food system is an important part of the world economy and society [1]. The annual grain harvest in the European Union is 300 million tons. Unfortunately, cereal grains are prone to infestation and contamination by warehouse pests during transport, storage and post-harvest processing in food and feed processing plants [2]. Grain losses in a storage account for up to 20% of stored products as a result of damage caused by insects [3,4], i.e., about 420 million tons of grain worldwide [5]. These losses vary greatly and depend mainly on the climate and species grown in a given region. With the increase in the average temperature, at which the pest development is much faster, the losses are greater, boosting to as much as 30% [6]. Traditional methods for controlling warehouse pests most often rely on the use of synthetic plant protection products. These control strategies pose the risk of food contamination with pesticide residues and the development of pest resistance to insecticides [7]. Feeding insects pose a particularly severe problem in the storage of cereals, as they lead to grain quality deterioration or grain damage [8]. The lesser grain borer (*Rhyzopertha dominica* F.) and the rice weevil (*Sitophilus oryzae* L.) are particularly difficult to control. In recent years, the first of the mentioned insects turned out to be a species capable of quick adaptation to unfavorable abiotic conditions and

significantly changed its habitat preferences in the conditions of our climatic zone [9]. An insect that prefers a warm climate, i.e., rice weevil, presents a growing threat due to global warming, and its occurrence and requirements are not sufficiently known and described in the temperate climate zone. In recent years, the progress of research on metagenomics has shown the symbiotic microbiome of insects to be closely related to the development of insects, metabolic changes in the body, resistance to unfavorable chemical factors and pathogens, and, to a large extent, to food digestion [10,11]. Therefore, the analysis of the composition, activity, and functions of the insect microbiome is a bioindicative method that allows for the precise determination of the impact of a factor introduced in the environment or breeding under laboratory conditions [10,12,13].

Bacterial endosymbionts of the insects' microbiome colonize the tissues of their hosts. They are common in the world of insects, and the relationship of the bacteria with the host may take the form of commensalism, mutualism, or parasitism. Parasitic bacteria manipulate the host to optimize the fulfillment of their own needs, whereas mutualistic bacteria provide metabolites or optimize the host's metabolic pathways (e.g., they participate in nitrogen circulation and can provide amino acids or vitamins), which is helpful or necessary in the life of the insect. Commensal microorganisms use the host periodically as a vector for migration purposes (e.g., to a plant to which it is a phytopathogen). The relationship between the bacterium and the host is variable and depends on the length of intercourse in the history of evolution, which is manifested by various intensities of interdependence and different coexistence strategies. Moreover, when microorganisms lose the benefit of symbiosis (or commensalism), they can become degraded or become pathogens to their host. There is a clear variability of conditions in the digestive tract of insects, i.e., the pH range, oxygen level, redox potential, availability of nutrients, and the functioning of the immune system. In its first section, the food-dependent microbiota is transferred and developed, which is largely dependent on the insect's developmental phase and environmental conditions. The middle section (lumen) of the digestive tract is colonized by symbiotic microbiota with specific functions influencing digestion and the immune system. The last section of the digestive tract consists of microorganisms that adhere to digesta and the epithelium. This group of microorganisms, largely composed of symbionts inhabiting previous sections of the digestive tract, can be transferred to eggs or exhibit coprophagic and trophallactic properties [14–19].

One of the important functions of symbionts in herbivorous insects is to provide the stored nutrients in conditions of starvation, as well as to break down plant biopolymers and inactivate the plant immune system, which seems to be a natural and necessary insect-symbiont interaction. Specialized groups of bacterial symbionts (obligatory symbionts) form in special intracellular structures called bacteriomes and are easily transferred vertically to the next generation of an insect species. Bacterial symbionts are transmitted both vertically and horizontally. Their composition determines, among others, insect reproduction potential, the redistribution of nutrients, and the stimulation of immunity against abiotic and biotic factors of the host [20]. Intestinal symbionts are an additional group of symbionts. They are responsible for the sharing and redistribution of nutrients and the cycles of elements, mainly nitrogen, across the digestive system; however, they also play a key role in detoxification processes and in overcoming the plant's immune system. Therefore, the range of plants that a given species of insects is able to feed on depends on this group of microbiota. Their transfer to other individuals is also vertical (e.g., coprotrophy) and horizontal [20,21]. The last group of microbiota that interact with insects lives outside the body, e.g., on food. Their task is to pre-break down plant tissue, provide simple substrates, and increase the amount of assimilable nitrogen. Of each of the mentioned groups, the most important ones, i.e., intracellular and gut microbiota, differ in the strategy of eliciting beneficial effects on the host. Unlike the gut microbiome, intracellular microbiota has strict functions with low flexibility, and their community shows a low gene pool due to evolutionary gene loss [20].

Typical examples of hosts associated with intercellular symbionts are the weevil species *Sitophilus* spp. Their gamma-protobacterium symbiont, *Sodalis pierantonius*, is able to provide rice weevil with essential vitamins such as pantothenic acid, riboflavin, and biotin. During the development of the larvae, bacteria are extensively regulated by the secretion system III genes and flagella to infect insect stem cells [22,23]. Moreover, the absence of these symbionts impairs the insect's immune system and its protection against pathogenic microorganisms [24].

However, it should be noted that a large part of the adverse impact of the environment on a pest does not have to be related to the insect microbiome. The disturbance in the functioning of the pest development may be direct, related to a direct dysfunction of the digestive system. The impairment of digestion was observed in Lepidoptera representatives due to the unfavorable parameters of wheat grain. Additional results were obtained by examining the life course and digestive activity of *Angoumois corn* moth α-amylase, *Sitotroga cerealella* Olivier (Lepidoptera: Gelechiidae). The researchers fed the pest with six wheat cultivars under controlled conditions and proved that one cultivar significantly impaired its development before reaching the imago stage. Moreover, significant differences were observed in the survival of immature insects, fertility, and body weight. The authors observed that the wheat cultivar with the highest 1000-grain weight had the best effect on the development and biometry of the larvae, and the highest amylolytic activity was observed in insects after feeding with this cultivar. In this work, the authors showed a strong correlation between the plant's ability to inhibit α-amylase activity and the impairment of insect pests (Lepidoptera) [25]. On the other hand, a clear connection was found between changes in the structure of the Lepidopteran intestinal microbiome and the body weight of the larvae and the development of the insects [26]. Research on understanding the function of the microbiome by sterilizing the insect microbiome is widespread in the case of mealworms [27–29], where insects treated with gentamicin, an antibiotic relatively safe for insects, lose their ability to digest feed.

So far, a huge number of bacterial taxa colonizing the gastrointestinal tract have been tentatively identified and their putative roles in the host organism have been described [30]. The aims of this study were to identify and characterize the microbiome of warehouse pests using the nanopore sequencing technique, sterilize the internal microbiome of warehouse pests and determine the lethal effect of different doses of the antibiotic, and describe the metabolic changes of the insects.

## 2. Materials and Methods

### 2.1. Mass Breeding

Adult species of the lesser grain borer (*R. dominica*) and rice weevil (*S. oryzae*) used in the experiment came from mass breeding conducted at the Department of Entomology, Phytopathology and Molecular Diagnostics, University of Warmia and Mazury in Olsztyn, Poland. The breeding was carried out under controlled conditions of temperature (30 °C) and humidity (70%) in a growth chamber (Sanyo MLR-350 H, Sanyo Electric Co., Ltd., Japan), with insect development, monitored and insect population regulated. Specimens of 1–2-day-old beetles were used in the experiment, which, after hatching in mass breeding, were intended for further experiments.

### 2.2. Experiment of Insect Exposure to an Antibiotic

The experiment investigated the development of rice weevil fed wheat wafer with the addition of gentamicin (Sigma Aldrich/MERCK, Germany) at various concentrations: 50 mg·g$^{-1}$, 40 mg·g$^{-1}$, 30 mg·g$^{-1}$, 20 mg·g$^{-1}$, and 10 mg·g$^{-1}$ per wafer. An antibiotic dose of a specific concentration (100 μL) was applied to 100 mg wheat wafer. The antibiotic was selected based on previous research on internal symbiotic microbiomes conducted on Coleoptera [27]. A wheat wafer supplemented with sterile demineralized water was used as a control. The wheat wafers prepared in this way were applied onto a Petri dish and 10 freshly hatched adults were used in this experiment. Each variant was made in four

series. The individuals were fed for seven days. In order to determine the survivability of the tested insects, the dead ones were counted after 0, 3, 5, and 7 days. After each counting, the insects were taken for further analysis.

### 2.3. Preparation of Insect Bacteriome

The internal body mass of *S. oryzae* and *R. dominica* was collected under a binocular in sterile conditions. Each subject was anesthetized at –80 °C for 5 min, after which the body surface was sterilized with 70% ethanol and rinsed thoroughly with sterile physiological saline (0.85% NaCl; demineralized water in the case of DNA isolation). Then, under contamination-limiting conditions, the chitinous (exoskeleton) parts of the insect and the head were removed. The prepared sterile samples were homogenized by grinding. Each homogenized sample consisted of three individuals from each treatment for microbiological analysis and biochemical test and total DNA isolation.

### 2.4. Microbiological Analysis

To determine the number of bacteria, the homogenized material was ten-fold diluted in sterile saline (NaCl 0.85%) and plated on TSA (tryptic soy agar, MERCK, Germany) medium. Incubation was performed at 28 °C for 48 h. Afterwards, the colony-forming units (CFU) were counted.

### 2.5. Biochemical Analysis

In order to determine the metabolic and enzymatic potential of the microbiome of the studied insects, three individuals were collected and the interior of each insect was dissected under sterile conditions, removing the exoskeleton. Prior to preparation, surface sterilization in 70% ethanol was performed, followed by three-fold rinsing in sterile demineralized water. The prepared samples were homogenized by grinding in sterile corundum (grain diameter < 0.1 mm) and then the homogenate was diluted in 1 mL of sterile physiological saline (NaCl 0.85%). Before the biochemical tests, a part of the prepared suspension was transferred to peptone water mixed with 1% TSB (Tryptic soy broth, Merck, Germany) and incubated overnight at 28 °C [10,11]. After overnight incubation, the pre-cultures were diluted to the appropriate optical density, depending on the test, and applied to the API NE and API ZYM test kits according to the instructions provided by the producer (Biomerieux, France). The assay kits were incubated at 28 °C for 24 h in the case of API 20 NE and 4 h in the case of API ZYM.

### 2.6. Isolation of DNA

Before starting the analysis, the insects were placed in a 1.5 mL Eppendorf tube containing 20 mg of corundum, approximately 0.2 mm in diameter, and 0.1 mL of a suspension buffer consisting of Tween 80 (0.005%) and saline (0.85% NaCl), and ground with a sterile micropestle. After pre-grinding, the obtained material of each sample was transferred to a 2 mL tube containing beads and a lysis buffer and homogenized in a TissueLyser LT homogenizer (Qiagen, Germany). Lysis was carried out for 5 min at the maximum speed (50 oscillations per sec). Further steps of the analysis were carried out following the instructions included with the isolation kit. After homogenization, the DNA was isolated from the collected material with a QIAamp PowerFecal DNA Kit (Qiagen, Germany) and concentrations for each sample were quantified by fluorometric quantitation using a Quantus ™ Fluorometer (Promega, Germany).

### 2.7. Nanopore Library Preparation, Sequencing, and Base-Calling

Sequencing was performed based on the "16S Barcoding Kit 1-24 (SQK-16S024)" [31]. The total DNA isolated from the insects was subjected to PCR with four different barcodes linked to primers covering the entire 16S rRNA region of bacteria included in the 16S Barcoding Kit 1-24 (SQK-16S024, Oxford Nanopore). The obtained amplicons were purified using Agencourt AMPure XP (Beckman Coulter). For library standardization, pooling was

performed to obtain a sample mixture containing 20 fmol of ~1500 bp amplicons in 10 μL. Then, 0.5 μL of a rapid sequencing adapter was added to 5 μL of the library. The Flongle flow cell was used for sequencing. Sequencing and base-calling were performed using the MinKnow software. The obtained data (~60,000 reads) were analyzed by the EPI2ME platform (Metrichor™ Ltd.) using WIMP (What's In My Pot) workflows.

*2.8. Statistical Analysis*

The number and species composition of the bacteria were analyzed. The species diversity of the analyzed bacteria was determined with the use of Simpson dominance (λ), Shannon diversity index (H'), and Pielou's evenness index (J'). The domination classes were determined according to previous work [32], where Eudominant accounts for more than 10%, Dominant, 5.01–10%, Subdominant, 2.01–5%, Rare, 1.1–2.0%, and Occasional, less than 1%, respectively.

The data from the microbiological analyses and survivability of the insects were tested firstly for normality of distribution (Shapiro–Wilk test) and homogeneity of variance (Levene's test), and then the differences between the results were determined by the Kruskal–Wallis test (at a significance level determined by Dunn's test with Bonferroni's correction). The results were presented as boxplots divided into quartiles with the median and mean. Statistical calculations of data from the test of survivability with different doses of the antibiotic were made based on the Kaplan–Meyer test. The relationships between observations were determined based on Pearson's correlation matrix, and agglomerative hierarchical clustering (AHC) based on the Ward's method. Heat maps of the biochemical properties were plotted based on a centralized and reduced data set.

# 3. Results

The analysis of α-diversity coefficients demonstrated that *S. oryzae* (Figure 1a) had a three-fold higher value of the dominance index, as well as greater diversity and uniformity, which was due to a much larger number of occasional taxa compared to *R. dominica* (Figure 1b). Based on the barcoding analysis of the full 16S rRNA region, six types of bacteria colonizing the microbiome of *S. oryzae* were determined with their proportion in the bacterial community of at least 0.5%.

On analyzing the general structure of the identified microbiota of the adult rice weevil form, the presence of two phyla with their proportion in the bacterial community at >0.5%, Proteobacteria 97.9% and Firmicutes 1.3%, was observed. Proteobacteria were almost completely dominated by the Pectobacteriaceae family with the participation of Enterobacteriaceae, while the Firmicutes structure included many species of Staphylococcus with a very low abundance. The results of the predicted species colonizing the insect bacteria demonstrated the complete predominance of *Sodalis pierantonius* and the relatively clear influence of *Escherichia coli* and the presence of *Sodalis praecaptivus*. It is surprising that it is *E. coli* bacteria that are eight times more numerous than *Enterobacter* spp., which form the typical insect gut microbiota (Figure 1a, Table 1).

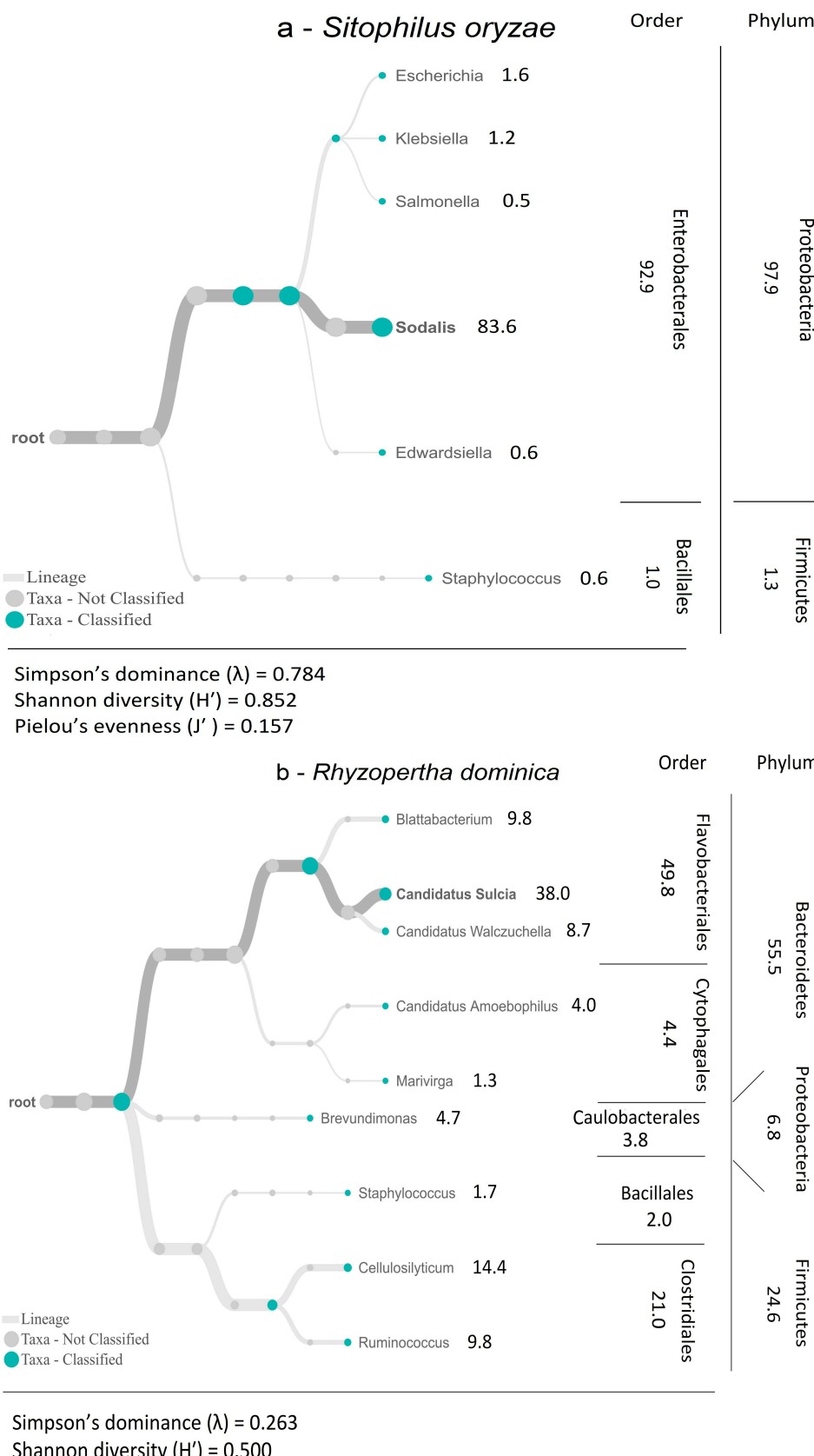

Simpson's dominance (λ) = 0.784
Shannon diversity (H') = 0.852
Pielou's evenness (J' ) = 0.157

Simpson's dominance (λ) = 0.263
Shannon diversity (H') = 0.500
Pielou's evenness (J') = 0.106

**Figure 1.** Results of nanopore 16S rRNA sequencing for the bacterial microbiome inhabiting the digestive system of (**a**) *S. oryzae* and (**b**) *R. dominica*.

**Table 1.** Taxonomic characteristics of the bacterial microbiome inhabiting the digestive system of *S. oryzae* and *R. dominica*.

| Pest | Species | Family | Proportion in Community (%) | Domination Class |
|---|---|---|---|---|
| *Sitophilus oryzae* | *Sodalis pierantonius* | Pectobacteriaceae | 81.9 | eudomination |
| | *Escherichia coli* | Enterobacteriaceae | 1.6 | rare |
| | *Sodalis praecaptivus* | Pectobacteriaceae | 0.7 | rare |
| | *Klebsiella pneumoniae* | Enterobacteriaceae | 0.6 | rare |
| | *Edwardsiella ictaluri* | Hafniaceae | 0.6 | rare |
| | *Salmonella enterica* | Enterobacteriaceae | 0.5 | rare |
| | *Proteus mirabilis* | Morganellaceae | 0.2 | occasionally |
| | *Enterobacter hormaechei* | Enterobacteriaceae | 0.2 | occasionally |
| | *Pectobacterium carotovorum* | Pectobacteriaceae | 0.2 | occasionally |
| | *Dickeya dianthicola* | Pectobacteriaceae | 0.2 | occasionally |
| | *Providencia rettgeri* | Morganellaceae | 0.2 | occasionally |
| *Rhyzophertha dominica* | *Sulcia muelleri* | incertae sedis | 30.7 | eudomination |
| | *Ruminococcus* sp. JE7A12 | Ruminococcaceae | 7.3 | dominantion |
| | *Walczuchella monophlebidarum* | incertae sedis | 7.0 | dominanton |
| | *Brevundimonas* sp. Bb-A | Caulobacteraceae | 3.8 | subdomination |
| | *Amoebophilus asiaticus* | Amoebophilaceae | 3.2 | subdomination |
| | *Blattabacterium punctulatus* | Blattabacteriaceae | 1.9 | rare |
| | *Blattabacterium cuenoti* | Blattabacteriaceae | 1.4 | rare |
| | *Marivirga tractuosa* | Marivirgaceae | 1.1 | rare |
| | *Chloracidobacterium thermophilum* | incertae sedis | 0.9 | occasionally |
| | *Staphylococcus pseudintermedius* | Staphylococcaceae | 0.8 | occasionally |
| | *Staphylococcus aureus* | Staphylococcaceae | 0.5 | occasionally |
| | *Weeksella virosa* | Weeksellaceae | 0.5 | occasionally |
| | *Clostridioides difficile* | Peptostreptococcaceae | 0.4 | occasionally |

The results of the nanopore sequencing analysis for *R. dominica* showed a much lower dominance of bacteriome taxa than in the case of *S. oryzae*. The phylum with a high proportion in the bacterial community includes Bacteroidetes, which is more than half of the bacteriome of the Flavobacteriales order (~50% of the bacteriome) consisting mainly of *Sulcia* sp. (38%), *Blattabacterium* sp., and *Walczuchella* sp. and the order Cytophagales with a small number of bacterial genera, represented most frequently by the genus *Amoebophilus* (4.0%). The phylum Firmicutes was also abundant with the proportion in the bacterial community of about 25%, almost completely dominated by Clostridiales with the *Cellulosiliticum* (~15%) and *Ruminococcus* (~10%) genera, and the order Bacillales (2.0%) represented mainly by bacteria of the genus *Staphylococcus*. The phylum Proteobacteria accounted for almost 7% of the bacterial community and was represented by *Brevundimonas* sp. (4.7%) and many less abundant taxa. The structure of the dominance classes of *R. dominica* differed from those of *S. oryzae*. *R. dominica* had eudominant species *Sulcia muelleri*; however, also dominant were representatives of *Ruminococcus* sp. and *Walczuchella monophlebidarum*, as well as two subdominants and three rare species, two of which belonged to *Blattabacterium* sp. (Table 1, Figure 1b).

The results of the microbiological analysis conducted for individuals feeding on feed with antibiotics showed a similar reduction in the number of bacteria in both species of insects. On the other hand, administered feed with antibiotics showed the stability of the microbiome structure and abundance of *R. dominica*, while the degeneration of the microbial community structure in the case of *S. oryzae* occurred in part of the variant. A significant reduction in the abundance of microorganisms was observed on the third feeding day at a dose of 30 mg·g$^{-1}$ and higher (Figure 2).

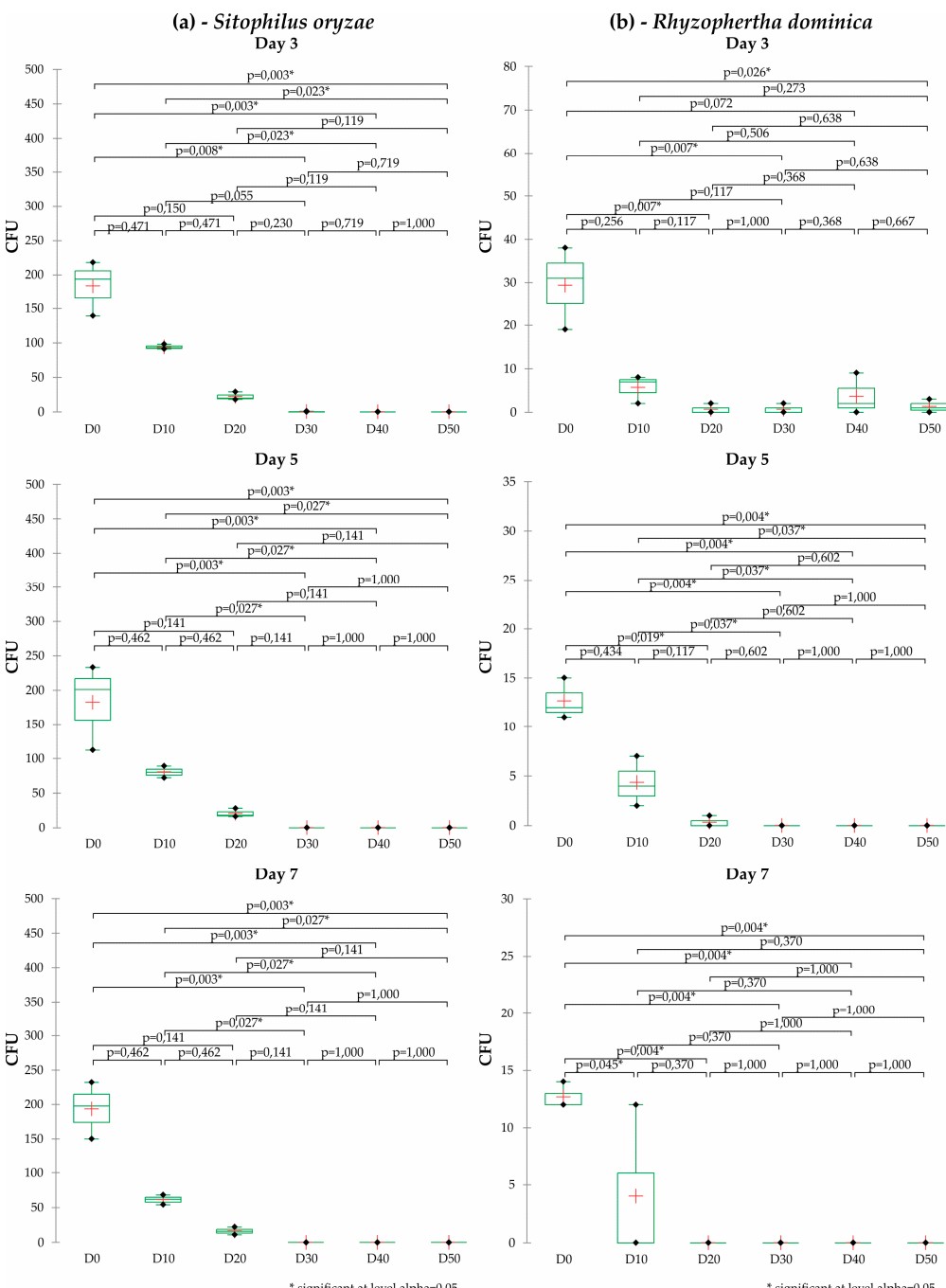

**Figure 2.** Boxplot showing changes in the number of bacteria (colony-forming units (CFU) in the consecutive days of culture on food contaminated with antibiotics (the initial number of bacteria for *S. oryzae* was 201 CFU and for *R. dominica* 40 CFU); D0: no antibiotic in the diet, D10 to D50: doses from 10 to 50 mg·g$^{-1}$ of the antibiotic in the feed.

Based on the Kaplan–Meyer test results, significant differences in the *S. oryzae* survival rate ($p < 0.05$) occurred in the starving insects (background) (survival rate 1.6 after seven days of culture) compared to the control, antibiotic doses of 10, 20, 30, and 40 mg·g$^{-1}$, and antibiotic doses of 0, 10, and 20 mg·g$^{-1}$ (survival rate ~0.94) versus 30, 40, and 50 mg·g$^{-1}$ (survival rate 0.19–0.46). It was observed that the survivability of the insects at an antibiotic dose of 30 mg·g$^{-1}$ did not differ from that observed at 40 mg·g$^{-1}$, but was significantly lower than the survival rate determined at a gentamicin dose of 50 mg·g$^{-1}$ (Figure 3a).

The survival rate of *R. dominica* after seven days of cultivation was ~0.83; nevertheless, no significant differences were shown in the Kaplan–Meyer test (Figure 3b).

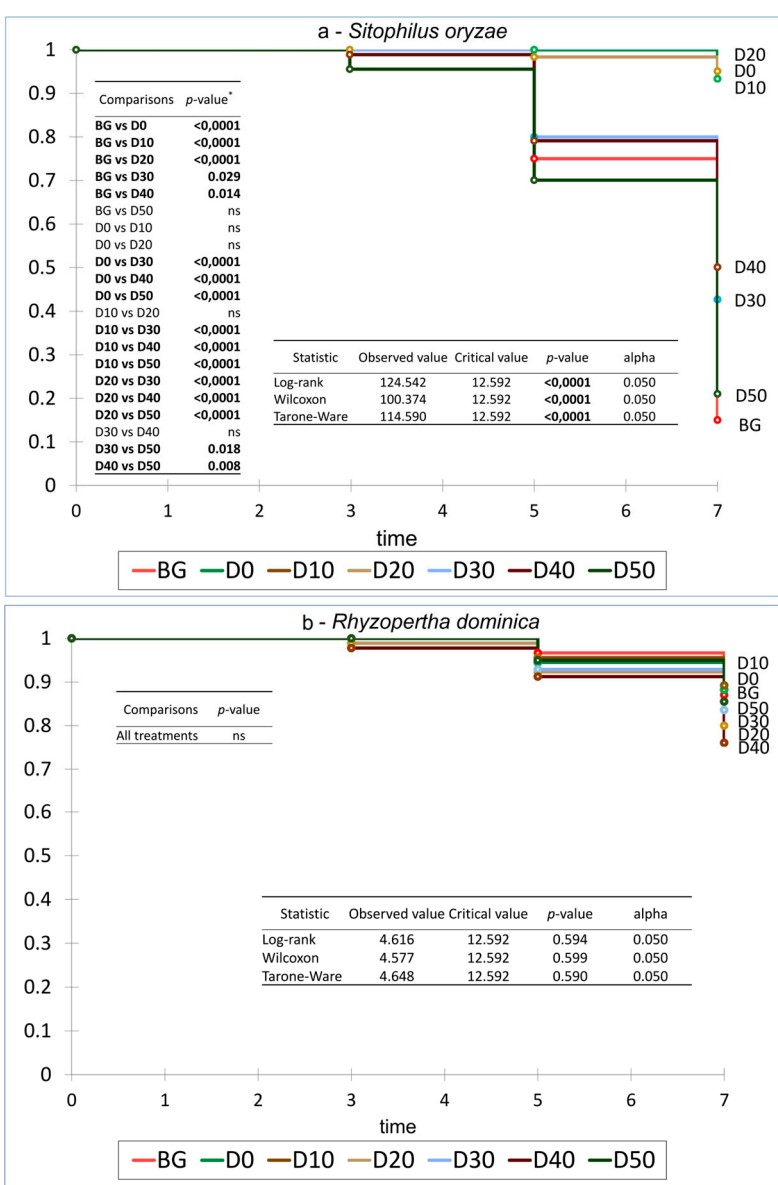

**Figure 3.** Kaplan–Meyer survival curve and *p*-value for a diet containing various doses of antibiotic and starving insects; BG: background (starving insects), D0: no antibiotic in the diet, D10 to D50: doses from 10 to 50 mg·g$^{-1}$ of the antibiotic in the feed.

The segregation of treatments (variables) on the AHC dendrogram for *S. oryzae* allowed two groups of very high dissimilarity to be distinguished. The first group consisted of the insects treated with the antibiotic doses from 0 to 20 mg·g$^{-1}$ with an almost low level of dissimilarity. The second group consisted of those treated with antibiotic doses from 30 to 50 mg·g$^{-1}$ with low within-group homogeneity. BG (background—starving insects) samples were included in the group of samples with a higher concentration of the antibiotic, which proves its similar effect on insect mortality (Figure 4a). In the case of *R. dominica*, a division into two groups was made for the lethal dose effect. The explicit segregation of results obtained in particular experimental series proved unsuccessful. Only the three samples of the variant with antibiotic does of 50 mg·g$^{-1}$ were categorized in group 2. Due to the extremely low dissimilarity for this dendrogram (~0.16), it should be assumed that there is no significant differentiation between the analyzed samples (Figure 4b).

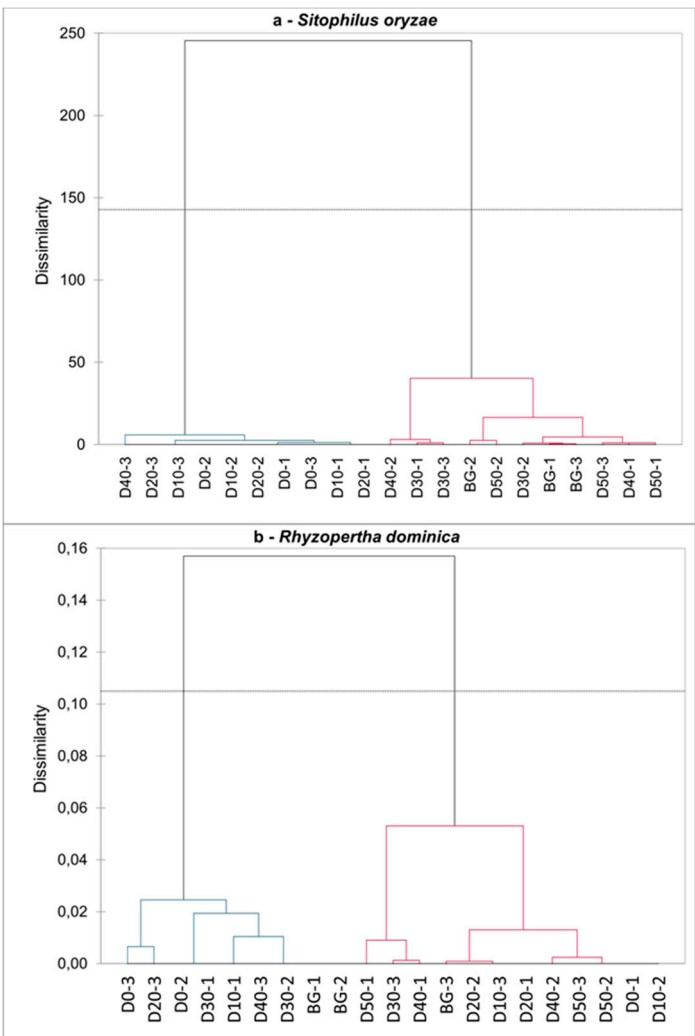

**Figure 4.** Agglomerative hierarchical clustering (AHC) dendrogram showing dissimilarity between samples based on lethal test values (**a**): *S. oryzae*, (**b**): *R. dominica*.

The results of the Pearson correlation test for *S. oryzae* showed that there were two nearly homogeneous groups of samples with a clear negative correlation. The first group consisted of each of the three replications for antibiotic doses from 0 to 20 mg·g$^{-1}$. The remaining high antibiotic doses (from 30 to 50 mg·g$^{-1}$) showed a negative correlation ($<-0.6$) with the aforementioned antibiotic doses that was found statistically significant in about half of the cases ($p > 0.05$; Figure 5a). The analysis of sterilization of the interior of *R. dominica* was also successful and at the antibiotic dose of 20 mg·g$^{-1}$, no bacteria were found on the media after five days of feeding on the contaminated feed. The sterile lesser grain borer showed a much better survival rate than the rice weevil (Figure 5b). Nevertheless, the individuals showed slightly weaker mobility, similar to the starving individuals (personal communication).

The grouping of the obtained biochemical profiles (left dendrogram) made it possible to distinguish two groups of variability. On the other hand, grouping the biochemical activity profiles of individual insects (treatments) allowed three separate groups to be distinguished (Figure 6). The first group (left clade on the dendrogram) consisted of live *S. oryzae* from the background and control groups. These lysates were characterized by high activity of α-fucosidase, β-glucosidase, and β-galactosidase as well as the utilization of D-maltose, L-arabinose, N-acetyl-glucosamine, potassium gluconate, malic acid, D-mannitol, trisodium citrate, 4-nitrophenyl-βD-galactopyroside, D-mannose, and reduction of nitrate to nitrite. Another group consisted of dissected guts from the dead background

*S. oryzae*, and live and dead *S. oryzae* fed on feed supplemented with the antibiotic. This group reduced in number and lost the activity or the ability to utilize substrates in at least 10 cases compared to the controls. The dead background specimens were characterized by high activity of α-fucosidase, β-galactosidase, 4-nitrophenyl-βD-galactopyranoside, lipase, β-glucuronidase, and naphthol-AS-BI-phosphohydrolase. A drastic reduction in the activity against the controls was observed in the case of esterase/lipase (C8), leucine arylamidase, β-glucosidase, and in the assimilation of potassium nitrate, D-mannitol, and D-glucose. Live insects receiving the feed with the antibiotic showed high activity of α-chymotrypsin and assimilation of D-mannitol and adipic acid. Seventeen traits did not change from the control, while about half of the observed traits showed a decline or loss of activity. In the case of dead individuals receiving the feed containing the antibiotic, only an increase in the utilization capacity of phenylacetic acid was observed compared to the control. The activity of the twenty traits did not change. An increase in the L-arginine utilization capacity was observed. Nevertheless, a decrease in enzymatic activity and the ability to utilize substrates was observed for: phosphatase alkaline, esterase/lipase (C8), esterase (C4), valine arylamidase, 4-nitrophenyl-βD-galactopyranoside, β-galactosidase, acid phosphatase, D-mannose, D-mannitol, malic acid, potassium gluconate, N-acetylglucosamine, L-arabinose, β-glucosidase, D-maltose, and the reduction of nitrate to nitrite (Figure 6).

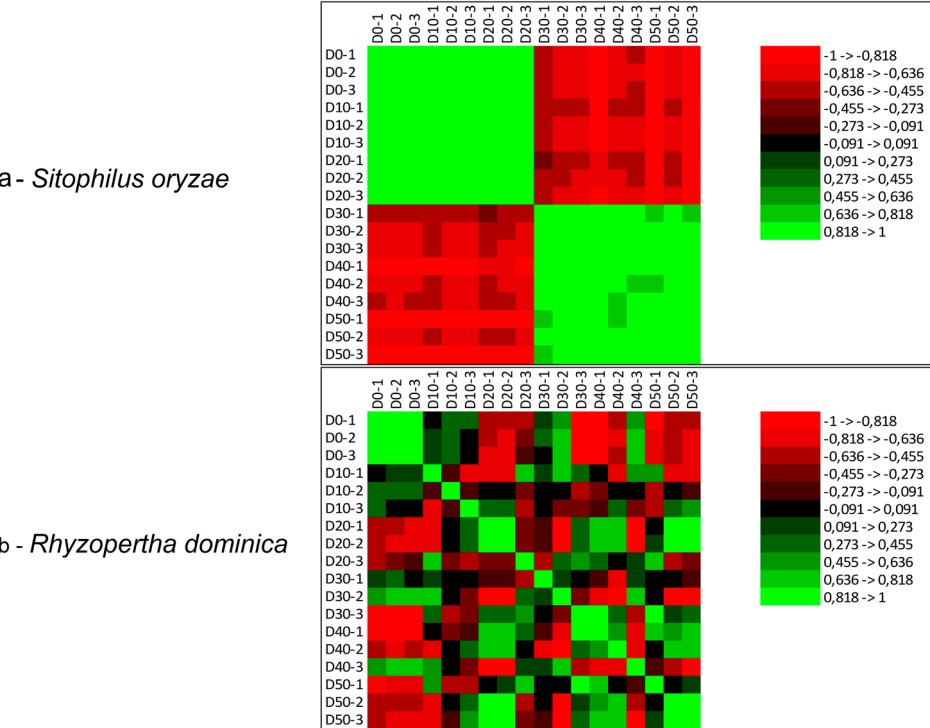

**Figure 5.** Heatmap based on cumulative correlation matrix (Pearson) of lethal test and bacteria CFU values ((**a**): *S. oryzae*, (**b**): *R. dominica*).

The last group consisted of all representatives of *R. dominica* included in this analysis. Regardless of the variants, all samples were characterized by the high activity of trypsin, cystine arylamidase, valine arylamidase, and protease, and the low activity of α-fucosidase, β-galactosidase, and β-glucuronidase. Guts prepared from control insects were only characterized by the lack of urease activity, while they had a high activity of traits common to this group mentioned above and α-glucosidase, α-chymotrypsin, esterase/lipase (C8), alkaline phosphatase, naphthol-AS-BI-phosphohydrolase, and adipic acid utilization. The activity of the remaining traits was at an average level. In the background individuals, only lipase activity (C14) increased compared to the controls. Twenty traits did not change,

while suppressed activity was observed for 17 traits. The greatest decrease was observed in chymotrypsin, esculin hydrolysis, acid phosphatase, and α-glucosidase activity, D-glucose fermentation, and adipic acid, phenylacetic acid, D-mannose, malic acid, potassium gluconate, N-acetyl-glucosamine, L-arabinose, and D-maltose utilization. In the variant with food containing the antibiotic, an increase in the activity against the control was observed in the case of leucine arylamidase, 4-nitrophenyl-βD-galactopyranoside, urease, and in the utilization of D-mannose, malic acid, L-arginine, and a reduction of nitrate to nitrite. No activity changes were observed for 25 traits, while suppressed activity was observed for naphthol-AS-BI-phosphohydrolase, alkaline phosphatase, esterase/lipase (C8), esterase (C4), α-glucosidase, and adipic acid utilization (Figure 6).

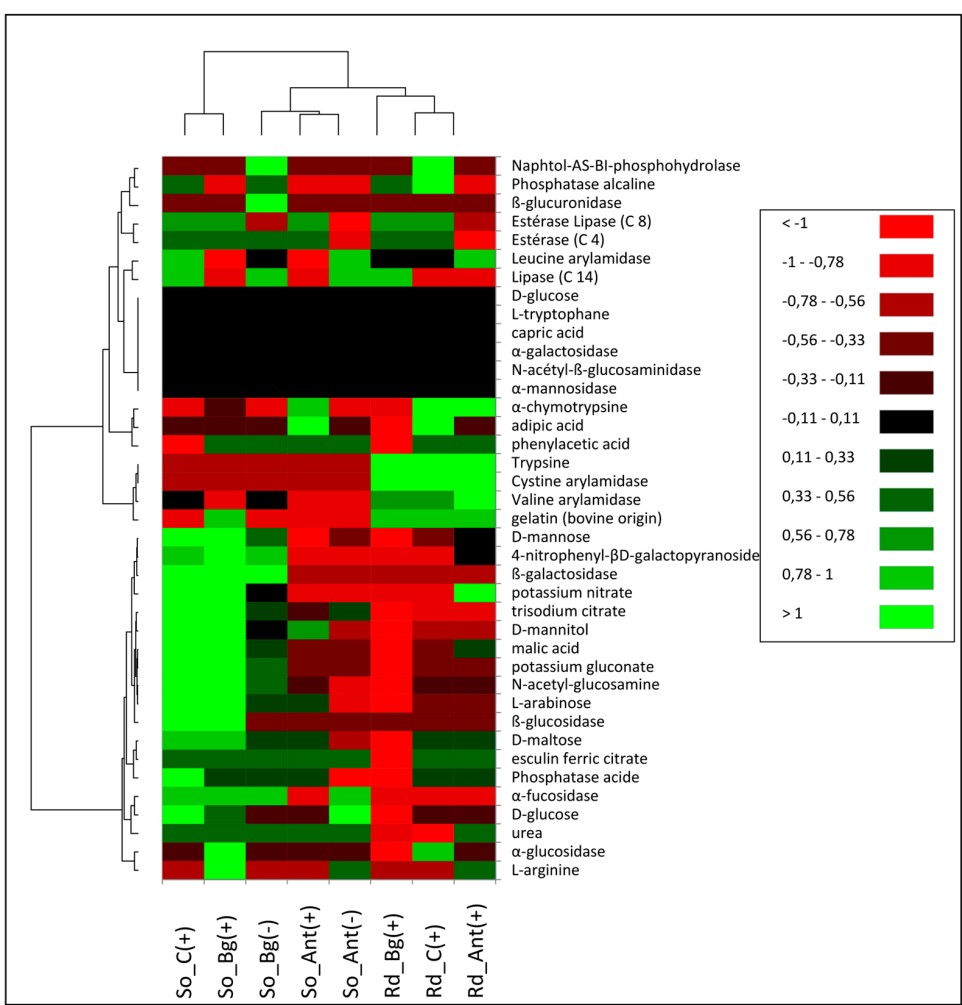

**Figure 6.** Heatmap showing the parameters of metabolism and enzymatic activity of the rice weevil microbiota and the lesser grain borer; So: *S. oryzae*, Rd: *R. dominica*, C: control, Bg: background, Ant: antibiotic, +: alive, –: dead.

## 4. Discussion

In this study, we confirmed that the loss of the symbiotic microbiota of the rice weevil caused a lethal effect, while the lesser grain borer, devoid of microbiota, showed a stable survival rate. It has also been proven that the lack of obligatory symbionts in the world of insects may cause, apart from the impaired utilization of nutrients, developmental impairment leading to death. Most of the research related to the role of the symbiotic microbiota concerns the gut microbiota and its important role in insect nutrition and immunology. A group of scientists [33] described the role of bacteria in the feeding of Asian long-horned beetle (*Anoplophora glabripennis*), which is an insect that feeds on the wood

of deciduous trees. This insect is incapable of degrading lignin and efficiently releasing bioavailable sugars. Nevertheless, the microbiome in the insect's intestine facilitates this process by allowing the insect to eat the sugars it has released. This group of scientists sequenced the microbiome of the digestive system of *A. glabripennis* and thus identified the genes of the microorganisms involved in the degradation of wood, including lignin. This group includes lactases, peroxidases, and β-esterases, as well as 36 families of glycosidic hydrolases, i.e., cellulases and xylanases, along with genes responsible for the bioavailability and increase in nutrient assimilation and their detoxification.

In addition to the function of the microbiota, it is also important to pay attention to the taxonomic groups of symbiotic bacteria associated with insects. Huang et al. [34] analyzed the intestinal microbiome of *Holotrichia parallela* larvae, whose larvae feed in the soil and cause severe crop yield losses. Based on 16S rRNA, ARDRA (amplified ribosomal DNA restriction analysis) and biochemical tests, a cellulolytic bacterial community was identified (Proteobacteria (70.05%), followed by Actinobacteria (24.15%), Firmicutes (4.35%) and Bacteroidetes (1.45%), which included the following genera: *Pseudomonas*, *Ochrobactrum*, *Rhizobium*, *Cellulosimicrobium*, *Microbacterium*, *Bacillus*, *Dyadobacter*, *Siphonobacter*, *Paracoccus*, *Kaistia*, *Devosia*, *Labrys*, *Ensifer*, *Variovorax*, *Shinella*, *Citrobacter* and *Stenotrophomonas*). Like in the present study, it was observed that the Proteobacteria-order bacteria were the prevailing symbiotic microorganisms, regardless of their function.

*Sodalis pierantnius* was the eudominant species colonizing the digestive tract of the rice weevils. Among the microbiota considered to be rare, but having a significant share in the intestinal community, the following were also classified: *E. coli*, *Sodalis praecaptivus*, *Klebsiella pneumoniae*, *Edwardsiella ictaluri*, and *Salmonella enterica*. The group of occasional species with a relatively high abundance included *Proteus mirabilis*, *Enterobacter hormaechei*, *Pectobacterium carotovorum*, *Dickeya dianthicola*, and *Providencia rettgeri*. These results clearly show that *Sodalis pierantonius* colonized the gastrointestinal tract as the eudominant bacterium. The remaining taxa were small in number and belonged to Enterobacteriaceae, typical representatives of the intestinal microbiota and, interestingly, other representatives of Pectobacteriaceae and *Proteus* sp., whose importance in the digestive system is unknown.

In turn, the *Sodalis* spp. genus bacteria have been identified as free-living bacteria, and a large group of species and strains were classified as insect symbionts. There was considerable genetic variation at the genomic level between free-living and symbiotic *Sodalis*. It was shown that symbionts had characteristic features such as a smaller genome size or a large number of pseudogenes. Nevertheless, the genome degeneration was not as great as for other symbionts. The abovementioned properties and the deletion-prone replication system indicate a recent period of symbiosis between these organisms and a potentially faster adaptation to the new host environment through adaptive mutations [35,36].

The research results obtained in this study suggest that the *Sodalis–Sitophilus* adaptive symbiotic system is most likely two-sided. This means that both the symbiotic bacterium and the host have abandoned some genome features useful for free-living organisms in promoting the dynamic adaptation and optimization of the mutual benefits of symbiosis. If *S. oryzae* did not lost its ability to function without becoming symbiotic, it is likely that the loss of the symbiont would significantly hinder its functioning; hence, it may be speculated that certain mechanisms responsible for nutrition, protection against pathogens, and/or survival in adverse environmental conditions would be impaired.

As indicated in the research by Tang et al. [37], *Nilaparvata lugens* bacterial symbionts are involved in the detoxification processes of this insect. The authors observed that high proportions of Wolbachia (including *Sodalis* spp.), *Arsenophonus*, *Acinetobacter*, and *Staphylococcus* bacteria were positively correlated with the activity of xenobiotic detoxification genes. In subsequent studies, a group of scientists [38] proved that bacteria of the genus *Acinetobacter* determine the detoxification of tea saponins in *Curculio chinensis*. In our research, next to eudominant *S. pierantonius*, we confirmed the presence of eudominant *S. praecaptivus* and *Staphylococcus* sp., which may be involved in detoxification processes and counteract negative environmental conditions. Survival tests showed that a weekly

fasting and feeding of the insects with food containing an antibiotic in a dose sterilizing the digestive system produced a comparable lethal effect of over 80%, with the highest mortality rate noted for the fasting individuals and the highest doses of gentamicin. On the other hand, the mortality rate of the individuals receiving the feed without the antibiotic did not exceed 10%.

Due to the subtle strategies of the host immune system in maintaining symbiosis with *S. pierantonius*, it appears that the lethal effect of fasting could entail a lack of nutrients for *S. oryzae*. The lack of symbionts could lead to dysfunction of the immune system, or, more surely, the transition of symbionts or other bacterial groups into parasitism or pathogenicity. The exact relationship of the immune system of the cereal weevil and Tsetse fly vs. *S. pierantonius* is described in Zaidman-Rémy et al. [39].

The lethal effect was also closely related to gastrointestinal sterilization. These observations prove that the lack of these bacteria impairs insect functions to a comparable extent to fasting. The most important function of the symbionts of the genus *Sodalis* is the production of cofactors, vitamins, and amino acids (tyrosine and phenylalanine). As shown by the scientific research of other authors, the lack of symbionts in the weevil leads to several distinct development anomalies [40–43].

Based on the results of nanopore sequencing analysis and the data in the KEGG database, we observed that *R. dominica* had bacteria serving different functions, the most important of which were carbon metabolism and the production of a large number of amino acids. Nevertheless, the presence of additional, unidentified symbionts raises questions about their function and prompts future metagenomic studies to take into account entire bacterial genomes. The above studies clearly show that *S. oryzae* has an obligatory symbiont and a very poor microbiome in terms of functional microorganisms, while a much larger number of dominant bacteria in *R. dominica* and the lack of their direct participation in the functioning of this pest may be the result of a much greater survival rate despite the sterilization of the bacteriome.

In this study, bacteria of the enphosymbiotic genera, such as *Sulcia* sp., *Cellulosilyticum* sp., *Walczuchella* sp., *Ruminococcus* sp., and *Blattabacterium* sp., were identified in the *R. dominica* bacteriome. In the study by Okude et al. [44], addressing the *R. dominica* bacteriome, the authors demonstrated that the main symbionts were pleomorphic and belong to Bacteroides. In addition, they included the following bacterial groups present in sucking-insect insects to this bacteriome: *Uzinaria*, *Brownia*, and *Walczuchella*, characteristic of scale insect, *Sulcia* sp., characteristic of a wider group of bugs, and *Blattabacterium* in omnivorous insects and xylophages, e.g., cockroaches [45]. The use of antibiotics in *Diaphorina citri* eliminated the insect symbionts *Candidatus* Profftella armature and *Wolbachia*, which suppressed the physiological process and metabolism of the insects [46].

In our research, we did not identify *Uzinaria* sp. or *Brownia* sp. Unfortunately, the pleomorphic symbiont *R. dominica* LC310894 [44] has not been identified due to the insufficient number of studies to date.

Another very important and confirmed discovery in these studies is the role of the key endosymbiotic bacteria in representatives of *Sitophilus* spp. As proven in the work of Vigneron et al. [47], until some point in adulthood, the amino acids phenylalanine and tyrosine, produced by the endosymbiotic *S. pierantonius*, are the precursors for the insect's production of DOPA (4-dihydroxyphenylalanine). This organic substance is necessary for the insect to stabilize and strengthen the structure of its exoskeleton.

As in previous studies involving pests [10,11], the symbiotic microbiome of insects partially determines the biochemical activity of the insect's body. Functional impairment, indirectly affecting the increasing mortality, was observed by increasing the depletion of the biochemical activity of *S. oryzae*. Owing to this, it was possible to identify features that constitute biomarkers of insects' functioning impairment. Based on the results presented on the heatmap, it can be seen that the activity of $\alpha$-chymotrypsin and esterase/lipase (C8) was lost in the dead individuals, and the ability to utilize mainly D-mannitol decreased. On the other hand, as it was observed, animals fed with the antibiotic, both still alive and dead,

lost a significant part of the features determining enzymatic activity, which resulted in their death in the following days of the experiment. In the case of *R. dominica*, it was shown that the most drastic effect on biochemical activity was triggered by starvation, while the antibiotic had a much weaker effect. This suggests that the microbiome of this pest plays a less significant functional (physiological and nutritional) role than that of *S. oryzae*. Therefore, individuals devoid of the microbiome were able to survive with a sterile digestive system. The starvation of *S. oryzae* at a key stage of exoskeleton development affects the lack of supply of amino acids of exogenous (feed) and semi-homogeneous (microbiome) origin, which indicates the high sensitivity of individuals to microbiome disorders, which indirectly contributes to their death.

## 5. Conclusions

Many countries are moving towards implementing sustainable solutions in plant protection. The initiatives in Europe include the "European Green Deal" or "Farm to Fork". They are aimed at drastically reducing the use of synthetic chemicals in agriculture, which have a strong environmental impact. The rice weevil and lesser grain borer are very important warehouse pests, and their reduction is strategic. Nevertheless, it will be difficult after the withdrawal of typical pesticides. The obligatory symbiotic nature of the *Sodalis pierantonius* bacterium at a key moment in the life of adult *Sitophilus* spp. opens up a new perspective for natural antibiotics and bactericides. Confirmation that certain insects enter into symbiotic interactions with the microbiota suggests the need for extensive testing of the pest's susceptibility to the loss of gut microbiota, which could open a new path in storage pest control strategies. Instead of using strong insecticides in storage and means of transport, non-synthetic substances could be applied to destroy the symbiont, which would hamper the development of the weevil caused, among others, by the lack of essential amino acids delivered by symbionts for the development of the host. Another strategy may be biological advances in new cereal varieties, leading to the natural production of bacteriostatic substances against *S. pierantonius*, which could help address the global problem of cereal storage losses caused by *Sitophilus* spp.

**Author Contributions:** Conceptualization, O.K., S.W.P. and M.N.; methodology, O.K. and S.W.P.; software, S.W.P. and O.K.; validation, O.K., S.W.P. and M.N.; formal analysis, O.K., S.W.P. and M.N.; investigation, O.K. and S.W.P.; resources, O.K. and S.W.P.; data curation, O.K. and S.W.P.; writing—original draft preparation, O.K., S.W.P. and M.N.; writing—review and editing, O.K., M.N. and S.W.P.; visualization, O.K., M.N. and S.W.P.; supervision, M.N. and S.W.P.; project administration, O.K.; funding acquisition, O.K., M.N. and S.W.P. All authors have read and agreed to the published version of the manuscript.

**Funding:** The results presented in this paper were obtained as a part of comprehensive study financed by the University of Warmia and Mazury in Olsztyn, Faculty of Agriculture and Forestry, Department of Entomology, Phytopathology and Molecular Diagnostics. This work was supported by a research project of the University of Warmia and Mazury in Olsztyn (no. 30.610.010-110 and 30.610.011-110). The study was supported by the Polish National Science Centre under the project no. UMO-2021/41/N/NZ9/00364.

**Institutional Review Board Statement:** Not applicable.

**Informed Consent Statement:** Not applicable.

**Data Availability Statement:** Metabarcoding data are available on the website (https://epi2me.nanoporetech.com/report-366053, accessed on 20 November 2022).

**Acknowledgments:** We would like to thank Bożena Kordan for her support in carrying out the research.

**Conflicts of Interest:** The authors declare no conflict of interest.

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
