# Peer review of "The Effect of Antibiotics on Bacteriome of Sitophilus oryzae and Rhyzopertha dominica as a Factor Determining the Success of Foraging: A Chance for Antibiotic Therapy in Grain Stores"

_applsci, doi:10.3390/app13031576_

Round 1

Reviewer 1 Report

The manuscript of Kosewska et al untitled “The effect of antibiotics on bacteriome of Sitophilus oryzae and Rhyzopertha dominica as a factor determining the success of foraging - a chance for antibiotic therapy in grain stores” focused on the evaluation of the internal bacteriome of S. oryzae and R. dominica before and after sterilization of internal microbiome with gentamicin and on the survival rate of both insects. I have already revised this manuscript when it was submitted a couple of months ago in another journal and it was already a good manuscript. The data seems sound and the manuscript is well written and in the scope of the journal. The minor modifications that requested in the previous journal have been considered, I do not see any other modification to request except one italic missing in the lines 310 and 315.

Author Response

We thank the reviewer for the careful review and thoughtful suggestion. 

The manuscript of Kosewska et al. untitled “The effect of antibiotics on bacteriome of Sitophilus oryzae and Rhyzopertha dominica as a factor determining the success of foraging - a chance for antibiotic therapy in grain stores” focused on the evaluation of the internal bacteriome of S. oryzae and R. dominica before and after sterilization of internal microbiome with gentamicin and on the survival rate of both insects. I have already revised this manuscript when it was submitted a couple of months ago in another journal and it was already a good manuscript. The data seems sound and the manuscript is well written and in the scope of the journal. The minor modifications that requested in the previous journal have been considered, I do not see any other modification to request except one italic missing in the lines 310 and 315. 

# Done: Italics were corrected in lines 317 and 322. 

Reviewer 2 Report

The manuscript original scientific data for bacteriome analysis

Manuscript needs more articles (with the publication years of last three years, especially articles published in 2022) to be cited throughout text

The subtitle of ‘’Main experiment’’ can be edited, please specify this title in detail

In 2.6 subtitle, please let us know about the authors used sterile pestle and mortar or not? And also, liquid nitrogen is used or not?

Author Response

We thank the reviewer for the careful review and thoughtful suggestion. 

Manuscript needs more articles (with the publication years of last three years, especially articles published in 2022) to be cited throughout text” 

# Latest papers have been added.in lines: 119-121, 424-426 and 463-465 

“The subtitle of ‘’Main experiment’’ can be edited, please specify this title in detail” 

# Section 2.2 was changed: Experiment of insect exposure to an antibiotic. (line 141) 

“In 2.6 subtitle, please let us know about the authors used sterile pestle and mortar or not? And also, liquid nitrogen is used or not?” 

# Done: see section 2.6, lines 182-185. 

Reviewer 3 Report

Based on the definition of the term DISINFECTION describing a process that kills many or all pathogenic microorganisms, except bacterial spores, ON INANIMATED OBJECTS (line 152) and STERILIZATION describing a process THAT EFFICIENTLY KILLS OR ELIMINATES ALL MICROORGANISMS, SUCH AS FUNGI, BACTERIA, VIRUSES and FORMS OF SPORES, except prions from a surface, equipment, food, medicine or biological culture medium (lines 9, 20, 120, 126, 298, 422, 432 and 447). The use of gentamicin is unlikely to produce such an effect in a pest, as they lack activity against anaerobes and most gram-positive bacteria, except many staphylococci; however, some gram-negative bacilli and staphylococci are resistant.

This error of using the term sterilization is repeated in line 167 when using 70% ethanol and stating that it promoted sterilization.

The document presents a proposal for pest control that goes against the global movement to control Antimicrobial resistance (AMR) who is a global health and development threat. 

Author Response

We thank the reviewer for the careful review and thoughtful suggestion. 

-„Based on the definition of the term DISINFECTION describing a process that kills many or all pathogenic microorganisms, except bacterial spores, ON INANIMATED OBJECTS"  

# Very old definition, used only in medicine and occupational health. In technological works, it has a more flexible meaning. In both definitions given by the reviewer, it is not possible to control microorganisms other than pathogens. 

Example 1- disinfection of infrastructure in food processing (we remove food spoilage microorganisms, but not necessarily pathogenic ones); 

Example  2- the use of antiseptics to disinfect the skin (i.e. living objects), which is a very common procedure. 

„The use of gentamicin is unlikely to produce such an effect in a pest, as they lack activity against anaerobes and most gram-positive bacteria, except many staphylococci; however, some gram-negative bacilli and staphylococci are resistant.” 

 - It says in the manuscript that we performed bacteriological cultures on rich media and the results were negative, which proves the sterility of the insect. In preliminary studies, we also used anaerobic culture media (Anaerocultu-MERCK), as well as culture media for molds and yeasts. In the case of yeast, single CFUs were indeed sporadically isolated, however, these were surface fungi and not of intrinsic origin. This is the subject of separate research and will not be included in this manuscript. Nevertheless, it is clear that a sterility check was performed, and this is largely confirmed by the tests in the manuscript. 

- Gentamicin has a positive effect on the microbiota of the insect digestive system, mainly because the large majority are G-. It is used in many scientific studies on insects. Extensive research on this topic has been described, e.g. by Yang et al. 2015 and that's what we relied on in this work. We have added an explanation in the text regarding the choice of antibiotic (line 145-147). 

“The document presents a proposal for pest control that goes against the global movement to control Antimicrobial resistance (AMR) who is a global health and development threat.  

- That was not the point of this work's perspectives. Sterilization of the microbiome with a synthetic antibiotic is a demonstration of the mechanism of action (no symbiotic bacteria - insect death). We wrote in the prospects that the next step is the development of natural antibiotics, or even varietal progress (e.g. polyphenols in grains), etc. for the biocontrol of symbiotic bacteria and, consequently, storage pests. 

Reviewer 4 Report

The manuscript submitted by Kosewska et al. merits to be published in Applied sciences as they carefully dissect the effect of the microbiome using antibiotic (gentamicin) and nanopore sequencing to characterize metabolic changes in two common grain insect pests, Sitophilus oryzae and Rhyzopertha dominica . The study is exceptionally well-written. I have very few comments about this paper as it very well designed and presented.

Major Comments:

My first comment is the choice of antibiotic; is gentamicin broad spectrum, does it preferentially kill Gram-negative or Gram-positive bacteria? The authors should comment on their choice of antibiotic and why they chose this one. 

My other comment is technical. When the authors write 30 mg.g-1, they should specify that it is not per g body weight of the insect but per gram wafer fed to the insect. I was confused by this notation.

Minor comments:

Also, in a follow up study, the authors should consider LC-MS to further validate their conclusions.

There is a typo 439, should be "data not shown" instead of "data not show".

Author Response

We thank the reviewer for the careful review and thoughtful suggestion. 

“My first comment is the choice of antibiotic; is gentamicin broad spectrum, does it preferentially kill Gram-negative or Gram-positive bacteria? The authors should comment on their choice of antibiotic and why they chose this one. “ 

- Gentamicin has a positive effect on the microbiota of the insect digestive system, mainly because the large majority are G-. It is used in many scientific studies on insects. Extensive research on this topic has been described, e.g. by Yang et al. 2015 and that's what we relied on in this work. 

- We have added an explanation in the text regarding the choice of the Line antibiotic (line 145-147). 

My other comment is technical. When the authors write 30 mg.g-1, they should specify that it is not per g body weight of the insect but per gram wafer fed to the insect. I was confused by this notation. 

# Corrected please see: Line 144. 

Reviewer 5 Report

Here, the authors test whether sterilizing the supposedly symbiotic microbiota of two warehouse pests, Sitophilus oryzae and Rhyzopertha dominica, could be implemented as a pest control strategy. They use gentamycin for sterilization purposes, and characterize the microbiota of interest as a whole both genetically (using 16S rRNA barcoding on a nanopore platform) and metabolically (using commercially available enzyme assay kits). Before the manuscript can be accepted for publication in Applied Sciences, the authors should integrate the answers to the following questions into their manuscript, add some extra figures, and apply the following corrections.

Major issues:
* How did the authors select gentamycin as the choice of antibiotic for their experiments?
* Why did not they test in advance a myriad of antibiotics against culturable bacteria from the midgut of warehouse pests in question?
* How did they determine the relevant doses of gentamycin?
* Two new figures, each showing an integration of the active pathways for the warehouse pest in question, would certainly be useful for functional metabolics purposes. The authors could possibly identify the affected pathways in their figures.

Minor issues:
* Line 141: "in various concentrations" should read "at various concentrations".
* Lines 307-333: species names should be written in italics
* Lines 375-376: either use "a group of scientists" or use "Huang et al. [33]".
* Overall: common names of the insects should start in lower-case letters.
* Overall: a thorough SPaG check is necessary.

Author Response

We thank the reviewer for the careful review and thoughtful suggestion. 

* How did the authors select gentamycin as the choice of antibiotic for their experiments? 

# - Gentamicin has a positive effect on the microbiota of the insect digestive system, mainly because the large majority are G-. It is used in many scientific studies on insects. Extensive research on this topic has been described, e.g. by Yang et al. 2015 and that's what we relied on in this work. 

- We have added an explanation in the text regarding the choice of the Line antibiotic (line 145-147). 

* Why did not they test in advance a myriad of antibiotics against culturable bacteria from the midgut of warehouse pests in question? 

#We tested a number of synthetic and natural antibiotics on selected storage pests. This data is part of an ongoing PhD process and will be published upon completion of the PhD. Nevertheless, there are a lot of studies testing various antibiotics on Coleoptera, which is why we chose the antibiotic used by many entomologist researchers, which is the antibiotic that works best and is safest for insects. 

* How did they determine the relevant doses of gentamycin? 

#Doses were obtained based on preliminary research as part of an ongoing PhD studying, and on previous research by Yang et al. 2015. 

* Two new figures, each showing an integration of the active pathways for the warehouse pest in question, would certainly be useful for functional metabolics purposes. The authors could possibly identify the affected pathways in their figures.  

#It should be borne in mind that this work is part of a complex study and a wide range of information will be available in subsequent studies. If we added additional information, we would not be able to answer the hypotheses, moreover, this manuscript is very extensive anyway, and adding new information would already be very troublesome both for the global study, but also for the reader. 

Minor issues: 
* Line 141: "in various concentrations" should read "at various concentrations". 

# Corrected, please see: Line 143 

* Lines 307-333: species names should be written in italics 

# Corrected, please see: Line 317-322 

* Lines 375-376: either use "a group of scientists" or use "Huang et al. [33]". 

# Corrected, please see: Line 382 

* Overall: common names of the insects should start in lower-case letters. 

# We made corrections throughout the manuscript. 

* Overall: a thorough SPaG check is necessary. 

# The manuscript has been reviewed by a native speaker and meets sufficient language requirements. 

Round 2

Reviewer 3 Report

Despite the indelicacy in the authors' responses, unfortunately, the authors were unable to answer the questions and much essential information.

I suggest you read the attached document, whose content and international regulations have resulted in the development of well-established post-harvest insect control methods and emerging technologies for the disinfection and detoxification of food grains.

Sincerely,

Author Response

Dear Reviewer,

My sincerest apologies for my indelicacy in my previous letter.

Nevertheless, I did not fully understand the previous allegations.
I also see that the terms concerned technical matters in the field of containment, not research methodology - in my opinion, in relation to this manuscript, we can only use issues related to the methodology of science because they are two separate matters.

According to the reviewer's statement, "Why develop something that is new and unregistered when standards are already developed". Probably not the way.

Let me remind you that this journal is called "Applied Sciences" and its task is to promote new methods, progress and improvement of existing solutions.

As for global regulations, of course they are developed but please note that they are insufficient, especially in some groups of countries that have their own.

I also forgot to add in the previous letter that all the words "disinfection" were standardized to "sterilization" so that there would be no more confusion.

Referring also to the ACTA case from a dozen or so years ago (people then went out into the street in white masks), one can see how different the laws of Europe, the United States, Japan and other countries and their approaches to international agreements are. Therefore, I believe that the regulations in force in the United States, the European Union, China, Russia and the Middle East should be approached individually.

I would like to point out that in the European Union, the "European Green Deal" is currently being introduced. This involves e.g. with a reduction in the use of synthetic pesticides. What does it mean for storage? The use of physical means, which, unfortunately, do not work always and everywhere. In addition, biological agents and biochemicals can be used (biocontrols and biological agents are already in use and are on sale and in use in many countries, and their importance, research, production, and sale will only now increase as an alternative to synthetic pesticides) and that's what they are for this research. In this work, of course, we used a synthetic antibiotic, because this is a prospective study to encourage other researchers to pay attention to the elimination of insect symbionts, which leads to the death of the insect. Future approaches, as we wrote in the paper, must be biological (bio-antibiotics or varietal progress - natural active substances present in the grain). Therefore, this work is strategic and I hope it will encourage many scientists to rely on it to develop safe bio-antibiotics to eliminate pests.

Even in the document in which Reviewnet sent the dates of updates and solutions in the field of bioeconomy and sustainable development, they will be more and more popularized, because they have already been approved, even in the countries of the European Union.

To update this regulation, there is a specific path to follow:
research is needed (e.g. this work and further ones in biocontrol aspects), an international project based on the proposal, creation of a blueprint for HACCP and a new technology line and implementation of the technology as a result.

Thank you very much for your review and I warmly encourage you to cooperate in future research and projects.

Best regards,

Sebastian Przemieniecki

Reviewer 5 Report

Unfortunately, the authors are unable to provide many pieces of essential information, which must be included in an original research article. They do this mainly because of their attempt to "salami-slice" one meaningful (unpuplished) paper, as mentioned in their author response letter (but disclosed from the manuscript itself). For more information about ethical concerns, please see Elsevier's factsheet on salami slicing (as attached here).

Author Response

Dear Reviewer,

My apologies, I probably forgot to mention this in the previous answer. We explained the use of antibiotics based on the literature (item 27 in the bibliography, and subsection 2.2).

I think that a number of tests, optimization or validation of methodologies are always carried out before the actual research.

Therefore, we believe that it is a developed method, and antibiotic testing has been available in scientific publications for at least
fifty years.
Going further, a very common "dilution method" is performed in classical microbiology. The methodological description of this method was first published in 1916. So this paper should be cited - and yet no one cites this paper.

I also suspect that the reviewer suspects the division of this work into slices.
The question is: horizontally or vertically?
Usually "slicing" refers to the horizontal division of research results. Example: we test 10 insects similar to each other. All the goals of the works are the same and we publish 10 separate works that differ only in the genre - yes, it is "cut into slices". When it comes to preliminary testing - it was true in our case, however, other insect species were for a different application, we also conduct research in the field of waste disposal by insects and bioindication. We wanted to save funds and we did it in one research package, if Reviewnet disagrees with this approach, I'm sorry, but unfortunately, due to poor funding of science, we are unable to conduct research in any other way. In this case, I ask for your understanding.
If the accusation of 'slicing' were that we would be testing the correct antibiotics on the insects used in these studies, i.e. antibiotics of natural origin (non-synthetic), or natural defense systems of cereals vs. symbiont, then the answer is simple: it is neither the purpose nor the form of this work. This is perspective study, it is intended to make people aware that nowadays (see: withdrawal from the use of synthetic pesticides, Green Deal program, the problem of carbon footprint, investing in the bioeconomy and sustainable development, etc.), this work is strategic, and I hope it will encourage many scientists to rely on it to develop safe bio-antibiotics to eliminate pests.

Thank you very much for your review and we encourage you to cooperate in future research projects.

Best wishes, Sebastian Przemieniecki